# Unlocking the Flexibility of District Heating Pipeline Energy Storage with Reinforcement Learning

**Ksenija Stepanovic** [1,*] , **Jichen Wu** [1,2], **Rob Everhardt** [2] **and Mathijs de Weerdt** [1]

1 Faculty of Electrical Engineering, Mathematics and Computer Sciences, Delft University of Technology, Van Mourik Broekmanweg 6, 2628 XE Delft, The Netherlands; jichen@flextechnologies.ai (J.W.); m.m.deWeerdt@tudelft.nl (M.d.W.)
2 Flex Technologies, Atoomweg 7, 3542 AA Utrecht, The Netherlands; rob@flextechnologies.ai
* Correspondence: k.stepanovic@tudelft.nl

**Abstract:** The integration of pipeline energy storage in the control of a district heating system can lead to profit gain, for example by adjusting the electricity production of a combined heat and power (CHP) unit to the fluctuating electricity price. The uncertainty from the environment, the computational complexity of an accurate model, and the scarcity of placed sensors in a district heating system make the operational use of pipeline energy storage challenging. A vast majority of previous works determined a control strategy by a decomposition of a mixed-integer nonlinear model and significant simplifications. To mitigate consequential stability, feasibility, and computational complexity challenges, we model CHP economic dispatch as a Markov decision process. We use a reinforcement learning (RL) algorithm to estimate the system's dynamics through interactions with the simulation environment. The RL approach is compared with a detailed nonlinear mathematical optimizer on day-ahead and real-time electricity markets and two district heating grid models. The proposed method achieves moderate profit impacted by environment stochasticity. The advantages of the RL approach are reflected in three aspects: stability, feasibility, and time scale flexibility. From this, it can be concluded that RL is a promising alternative for real-time control of complex, nonlinear industrial systems.

**Keywords:** 4th generation district heating; combined heat and power economic dispatch; Markov decision process; mixed-integer nonlinear program; pipeline energy storage; *Q*-learning

## 1. Introduction

Energy storage systems are crucial for providing flexibility to the power systems [1]. An important source of flexibility is the thermal energy storage present in the district heating network (DHN), which is found in the thermal inertia property of the network itself [2]. The use of this flexibility for balancing power is an important element in 4th generation district heating [3]. Utilization of the pipeline energy storage decouples heat and electricity production of the combined heat and power (CHP) unit while meeting consumers' heat demand [4]. Decoupling of heat and electricity enables CHP production to exploit fluctuations in electricity prices and renewable energy sources (RESs), both for profit gain and higher utilization of RESs. The profit gain is achieved by trading electricity with an external electricity grid at favorable moments. Higher integration of power produced by RESs is reached by lowering CHP electricity production with respect to heat production and consumers' heat demand [5].

Electricity trading is performed in a number of energy markets, most importantly day-ahead and intraday (real-time) markets. In the day-ahead market, production facilities are scheduled to operate during each hour of the following day [6]. In the intraday market, participants trade electricity continuously with the delivery on the same day. Electricity can be traded up to five minutes before the delivery, and through hourly, half-hourly,

and quarter-hourly contracts. An increase of renewable energy production results in the growing importance of the real-time electricity market, and trading on it [2]. Not requiring any investment into dedicated storage tanks, pipeline energy storage allows using additional flexibility at the minimum cost [7].

Optimal scheduling of CHP production units is usually done without considering grid dynamics [8]. The inclusion of the district heating pipeline introduces a complex nonlinear description of the non-stationary temperature propagation processes through the network. The main challenges in the control of the district heating system are:

1. Environment uncertainty
2. Modeling uncertainty
3. Scarcity of placed sensors

The environment uncertainty originates from prediction uncertainties of heat demand, electricity prices, and/or RESs. The control strategy should cope with these uncertainties in an online manner. Common uncertainties in modeling district heating systems arise due to simplifications and assumptions about the physics, exclusion of some energy systems due to the modeling effort, and unknown or wrongly estimated model parameters [9]. Modeling uncertainty can result in infeasibility when solutions of the optimization are evaluated on the actual grid.

The sensors required to determine the state of the thermal network are scarce, as the pipeline is an enclosed underground system. As the shortage of delivered heat will manifest as a low-pressure difference at the consumer furthest away from the production plant, a pressure sensor is commonly placed there [10]. The deficiency of sensors imposes an additional challenge in inferring the state of the system and reducing modeling uncertainty.

### 1.1. Related Work

Previous works on the integration of heating grid dynamics into the scheduling of CHP plants tried to address the above challenges. A number of studies start from detailed, nonlinear physics models of the heating grid, including integer and continuous decision variables, to reduce modeling uncertainty. However, there are no known algorithms with robust and predictable performance for solving the resulting mixed-integer nonlinear programs (MINLP) [11].

A decomposition approach suggests a division of the MINLP on subproblems and their iterative optimization. Li et al. [12] divided the model of the heat transfer through the pipeline into two subproblems to decouple integer time delays from continuous decision variables. Runvik et al. [11] also partitioned the model on two subproblems to address complex dynamics of the grid and on/off status of CHP plants. When subproblems are still nonlinear models, and when there is a high number of shared variables in these subproblems, the coupling variables may not converge [13]. The long computation time and large-scale dimensionality of the model are restricting factors for use in practice.

A great number of research contributions simplify grid topology under various assumptions to strengthen guarantees on stability, convergence, and facilitate online application [14–16]. Merkert et al. [7] established the direct link between supply temperature and heat stored in the pipeline under the assumption of constant transport times from heat source to consumer. In [15], heat loss is modeled as a function of the pipe length and surface, disregarding the time dimension. Verrilli et al. [17] formulated a large-scale district heating system as a mixed-integer linear program (MILP) by excluding the model of the district heating pipeline. This simplification enabled a receding horizon optimization which enhances robustness to forecast errors of the heat demand. The quality of such approximations depends on the skills and experience of modelers and the availability of information on the relevant parameters [18]. If the original model is relaxed, the optimization output may result in constraint violations when evaluated on a more detailed model of a simulator or in practice, resulting in unforeseen modifications to the plan and the consequential costs.

Recently, reinforcement learning (RL) algorithms started being applied to district heating systems control to account for environment and modeling uncertainties. Learning

the optimal set of actions with RL requires large number of interactions with the simulation or real-world environment of the process. In [19], a distributed proximal policy optimization algorithm determined the control strategy for multiple production units to maximize the profit. Zhang et al. [20] used the proximal policy optimization algorithm to determine the optimal wind power conversion ratio. Claessens et al. [21] successfully applied fitted Q-iteration for the optimal control of thermostatically controlled loads connected to the DHN. Authors motivate the use of RL by the potential to deal with the complex dynamics of nonlinear systems, the high dimensionality of the problem, the adaptability to various operation scenarios without recalculation, and the short reaction times, allowing their use in online control.

The current challenge is to address the modeling and environment uncertainty under sensor placement scarcity. An algorithm is needed with a suitable approximation of the complex elements in the DHN that mitigates constraint violations (1st criteria) while improving profit gain (2nd criteria). The control strategy should operate stably (3rd criteria) over changing heat demand and be applicable in real time (4th criteria). The limitation of existing DHN controllers is the exploration of only one or two of these criteria.

*1.2. Main Contributions*

In this work, we develop a control strategy that relies on RL and explore its potential in efficient, stable, safe, and real-time control. The RL algorithm learns the system's dynamics through interactions with the simulation environment. To the best of our knowledge, there exist no previous applications of RL with pipeline energy storage considering temperature–mass flow dynamics.

The contributions of this paper are summarized as follows:

- Modeling of CHP economic dispatch with DHN pipeline storage as a Markov decision process:
  - The construction and performance comparison of the RL with a full state space and a partial state space created from the realistically available sensory information.
  - The creation of the reward function incorporating a system's limitations and safety guarantees.
- The RL approach is compared with an exhaustive mathematical optimizer [12] on a half-year dataset for trading on the day-ahead and real-time electricity market, and it showed better stability guarantees, higher feasibility when evaluated with the simulator, and time scale flexibility, while making moderate profit gain for shorter pipes.

The rest of this paper is organized as follows: Section 2 describes the mathematical model, optimization algorithm, creation of upper bound on profit, and basic control strategy. Section 3 describes and motivates the RL algorithm and the development of its components—action space, full and partial state space, and reward function. Section 4 introduces and motivates grid setting and parameter values. Section 5 provides an overview of the simulation environment used for the RL algorithm training and algorithms' evaluation. Section 6 presents case studies and discusses the results. Finally, Section 7 concludes the article.

## 2. Mathematical Modeling and Optimization of the District Heating System

In this part, a representative state-of-the-art mathematical model of the district heating system components, an objective function formulation, and an optimization algorithm are described. This section identifies the reason for using the described mathematical model as a benchmark and also its challenges and limitations. Moreover, the mathematical model specifies safety requirements that our newly developed RL control strategy should satisfy. At the end of this section, we define a model of an upper bound for profit and a basic control strategy used for comparison of the algorithms.

### 2.1. Mathematical Model of the District Heating System

A district heating system consists of a heat station (HS), a heat exchange station (HES), and district heating pipelines.

The main elements of the HS are a CHP unit and a water pump. The CHP unit simultaneously produces heat and electricity defined by the operating region in Figure 1. The drawn operating region is commonly used in the literature [19,22,23].

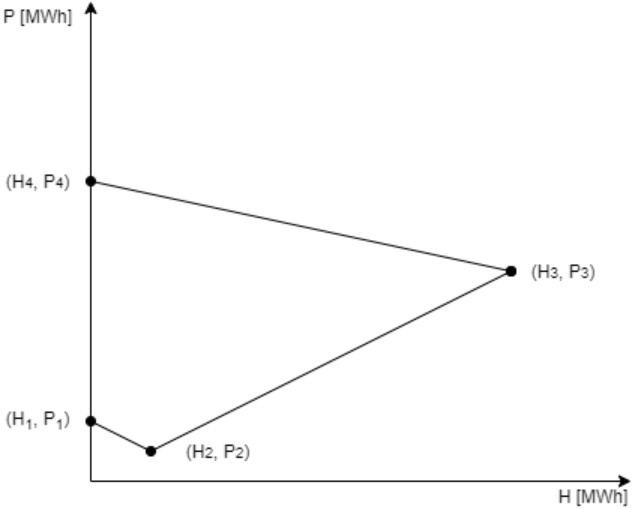

**Figure 1.** A combined heat and power unit can produce any combination of heat (horizontal axis) and power (vertical axis) within the region spanned by the four corner points.

The produced heat $h_t$ and electricity $p_t$ at time-step $t \in T$ is a convex combination of $N_P$ extreme operating points of heat $H_i$ and power $P_i$ [24]:

$$h_t = \sum_{i=1}^{N_P} \alpha_{t,i} H_i, \quad p_t = \sum_{i=1}^{N_P} \alpha_{t,i} P_i, \quad t \in T \tag{1}$$

where

$$\sum_{i=1}^{N_P} \alpha_{t,i} = 1, \quad 0 \leqslant \alpha_{t,i} \leqslant 1, \quad i = 1, \ldots, N_P, \quad t \in T \tag{2}$$

The objective is the maximization of profit $F$, i.e., the sum of the cost $a_0 h_t$ and $a_1 p_t$ for producing heat and electricity, respectively, and the gain $c_t p_t$ from the sale of electricity to the external grid over optimization horizon $T$:

$$F_t = -a_0 h_t - a_1 p_t + c_t p_t, \quad F = \sum_{t=1}^{T} F_t \tag{3}$$

For inspecting the significance of the district heating pipeline on profit gain, we also define the linear program (LP) for a CHP economic dispatch without considering district heating grid dynamics. The linear model is defined with Equations (1)–(3), and with an additional constraint that the produced heat should meet the consumer's heat demand, $h_t = Q_t$.

The produced heat is equal to the product of mass flow $\dot{m}_{s,t}$ and a temperature difference between the supply $\tau_{s,t}^{in}$ and return network $\tau_{r,t}^{out}$ at the HS:

$$h_t = C \dot{m}_{s,t} (\tau_{s,t}^{in} - \tau_{r,t}^{out}), \quad t \in T \tag{4}$$

The temperature of the supply network at the HS is also called the supply network inlet temperature (that explains the upper index *in* in symbol $\tau_{s,t}^{in}$).

The supply network inlet temperature is bounded. The lower bound $\tau_s^{min}$ provides a stronger guarantee for the consumer's demand satisfaction, while the upper bound $\tau_s^{max}$ is the system's safety requirement:

$$\tau_s^{min} \leqslant \tau_{s,t}^{in} \leqslant \tau_s^{max}, \quad t \in T \tag{5}$$

Delivered heat $h_t^c$ is proportional to the mass flow and temperature difference between supply $\tau_{s,t}^{out}$ and return network $\tau_{r,t}^{in}$ at the HES. The delivered heat is equal to the consumer's heat demand:

$$h_t^c = C\dot{m}_{s,t}(\tau_{s,t}^{out} - \tau_{r,t}^{in}), \quad Q_t = h_t^c, \quad t \in T \tag{6}$$

The return temperature at the HES is bounded. The lower bound $\tau_r^{hes,min}$ prevents significant lowering of the temperature in the return network and, subsequently, the necessity for high supply network inlet temperature:

$$\tau_r^{hes,min} \leqslant \tau_{r,t}^{in}, \quad t \in T \tag{7}$$

The mass flow rate in the supply and return network is smaller than the maximum value $\dot{m}^{max}$. Otherwise, breakage of the pipe might occur. Due to the network structure we use (more details in Section 4), mass flows in the supply and return network are equal:

$$0 \leqslant \dot{m}_{s,t} \leqslant \dot{m}^{max}, \quad t \in T \tag{8}$$

Further equations for the power consumed by the water pump, a feasible region of the water pump power, a constraint on the pressure of the heat load, a continuity of the mass flow, and the pressure loss [25] are implemented in the mathematical model. More details on the mathematical model are provided in [12].

The above-described mathematical model is the most detailed model of heat transfer processes in a DHN we have found. Therefore, we use this model to benchmark our novel RL approach. This mathematical model is a large-scale mixed-integer nonlinear program, MINLP. The complexity of the model represents a major challenge for solving it. Currently, there are no known exact algorithms with predictable and robust performance for solving MINLP [11]. In the next section, we describe the decomposition approach, which is one of the possible indirect approaches used for solving MINLP.

### 2.2. Solving the Mathematical Model

In this section, we describe how to use a state-of-the-art mathematical optimizer to solve the MINLP model of the heat transfer processes in the DHN, which was introduced above. MINLP contains integer and continuous variables as well as linear and nonlinear constraints. Complex constraints make it difficult to solve the model with off-the-shelf solvers. An alternative is the decomposition approach. Decomposition provides a general framework where one splits the original problem into smaller subproblems and combines their solutions into an easier global problem [26].

The complexity in the DHN originates from the temperature propagation process through the pipeline. A node method models transient temperatures through the district heating pipeline [27]. The node method is the simplification of a more detailed and more accurate but computationally more expensive element method [28]. The principle of the node method is the estimation of time delays needed for the water mass to outflow the pipeline. Using estimated time delays, the temperature at the pipe's outlet is approximated with rescaled inlet temperatures. Modeling of the heat loss in the node method is enhanced in [29].

When solving the model of the district heating system, challenges are nonlinear relations between mass flow, time delays, and outlet temperatures in the node method. To address these complex integer and nonlinear dependencies, the model is divided into two simpler submodels [12]. Still, a submodel is a nonlinear program. Submodels are iteratively solved until the convergence of shared variables or reaching the time limit. The

output of the model includes the heat and electricity production pairs $(h_t, p_t)$, $t = 1, \ldots, T$. This approach to solving the MINLP introduced in the previous section will be the main benchmark for evaluation.

### 2.3. Model of a Profit Upper Bound

Another benchmark is the following upper bound on the profit. The flexibility originating from the thermal inertia in the district heating system enables greater profit gain compared to the CHP economic dispatch without grid dynamics. The upper bound $\overline{F}$ is developed under the assumption of a maximum possible time delay in the electricity dispatch to the external grid. This assumption relaxes realistic constraints on time-delay values in the grid. Therefore, it enables the sale of electricity for the same or higher price compared to the solution of the DHN model with grid dynamics. Produced heat and electricity satisfy Equations (1) and (2). Equation (3) is modified to incorporate maximum flexibility in the electricity dispatch:

$$\overline{F}_t = \max_{k=t,\ldots,T} -a_0 h_t - a_1 p_t + c_k p_t, \quad \overline{F} = \sum_{t=1}^{T} \overline{F}_t \tag{9}$$

This model is easy to compute, and the resulting profit is guaranteed to be higher than any feasible schedule. It will be used to establish the quality of the results for performance criteria.

### 2.4. Model of a Basic Control Strategy

To further evaluate the performance of the mathematical optimizer and the RL algorithm, we calculate the profit obtained by following a basic control strategy. The basic control strategy is derived by the simulator (more details on the simulator in Section 5). In the basic control strategy, an input to the simulator is the constant supply network inlet temperature of 90 °C. The input differs from previously described mathematical models in Sections 2.1 and 2.3, where the main decision variables are heat and electricity production defined by operating region from Figure 1. Based on the constant supply inlet temperature as input and heat demand, the mass flow required to satisfy the heat demand is determined by the heat exchange process of the simulator. The produced heat $h_t$ is calculated by Equation (3) and produced electricity $p_t$ following the operating region from Figure 1. The cost $F_t$ is defined by Equation (4). However, specifying the basic control strategy in this way has a limitation. When the produced heat is calculated based on the supply inlet temperature, it can exceed the feasible operating region defined in Figure 1.

## 3. Formulation of a Reinforcement Learning Framework for the District Heating System Control

As concluded from the related work, the three main challenges in district heating network control are the inaccuracy of simpler models, uncertainty from the environment, and limited availability of real-time information. In this section, we present the choices to be made when addressing the DHN control problem with reinforcement learning and discuss and reason about the viability of different elements of RL algorithm—the action space, the state space, and the reward signal.

To learn a good mapping from states to actions, RL requires a large amount of interactions with the simulation environment. Therefore, we developed the simulator of physics processes in the district heating system (more details in Section 5). Formulation of the DHN control as RL framework enables us to avoid the introduction of inaccuracies such as in mathematical models by learning the model's characteristics through interactions with a simulation environment. Additional advantages of RL are a highly encapsulated model, adaptability to different operation scenarios, and time scale flexibility in uncertain scenarios.

In an RL approach, the RL environment is often formalized as a Markov decision process (MDP), which is described by a 4-tuple $(S, A, P, R)$. At each time-step $t$, the agent interacting with the environment observes a state $s_t \in S$ and selects an action $a_t \in A$,

which determines the reward $R_t \sim R(s_t, a_t)$ for that action in that state, and the next state $s_{t+1} \sim P(s_t, a_t)$. The $Q$-learning algorithm [30] is a basic RL algorithm. In this algorithm, the value of executing an action in the given state, called the $Q$-value, is estimated. $Q$-values are learned through interactions with the environment by updating the current $Q$-value approximate toward the received reward $R_t$ and the approximated utility of the next state $Q(s_{t+1}, a_{t+1})$ [31]:

$$Q(s_t, a_t) = Q(s_t, a_t) + \alpha(R_t + \gamma \max_{a_{t+1}} Q(s_{t+1}, a_{t+1}) - Q(s_t, a_t)), \quad Q(s_0, a_0) = 0, \qquad (10)$$

where $\alpha$ determines the learning rate and $\gamma$ discounts future reward. The agent's final goal is to maximize the cumulative reward, the sum of all received rewards. To achieve this goal, the RL agent needs to obtain an accurate representation of the environment.

When the agent–environment interaction breaks naturally into subsequences (episodes), the task is episodic. Each episode ends in the specific state, which is called the terminal state. The ending of an episode is followed by the reset to a standard start state. In an episodic task, the agent goal is to maximize the cumulative reward collected through an episode, $G_t = R_{t+1} + R_{t+2} + \cdots + R_T$, where $T$ denotes the final step of the episode [32].

Commonly in the district heating grid, the pressure sensor is placed at the furthest consumer. To inspect the importance of sensor placement on the state reconstruction, we consider the desirable case where the agent receives an observation $o_t$ of the temperature at the inlet $\tilde{\tau}_{s,t}^{in}$ and outlet of the supply pipe $\tilde{\tau}_{s,t}^{out}$, and the mass flow $\tilde{m}_{s,t}$ at the time-step $t$. A partially observable Markov decision process (POMDP) better captures the dynamics of this environment by explicitly acknowledging that the sensor inputs received by the agent are only partial indications of the underlying system state. A POMDP can be described as a 6-tuple $(S, A, P, R, \Omega, O)$. $S, A, P$, and $R$ are states, actions, transitions, and rewards as in the MDP, but instead of receiving an environment state, the agent receives the observation $o \in \Omega$ generated from the underlying system state according to the distribution $o \sim O(s)$. Given the observations, the agent updates its belief (probability distribution) about the current state. An MDP can be seen as the special case of POMDP where the state is observed completely.

We next describe the action space, state space, and the reward signal more precisely.

### 3.1. Action Space

The control variables are produced heat and electricity according to the operating region in Figure 1. Because tabular $Q$-learning is a discrete reinforcement learning algorithm, the action space is formed of heat and electricity production pairs, which are taken at an equal distance from the edges in Figure 1:

$$a_t = (h_t, p_t) \in (h_i, p_i), \quad t \in T, \quad i = 1, \ldots, N_A \qquad (11)$$

The number of such points, $N_A$, is a parameter of the approach which can be chosen depending on the size of the Q-matrix and training time of the $Q$-learning algorithm.

### 3.2. State Space

The performance of the RL agent heavily relies on the model chosen to represent the state. The modeled state captures the information available to the agent about its environment. Constructing, designing, and changing a state is a challenge, as representative states are not necessarily the raw observations from the environment [32]. The first part of the challenge is determining which environment observations to include and whether past observations are part of the state. If the state captures partial or modified information from the simulation environment, that introduces randomness in the decision-making process of the RL agent (environment stochasticity). Then, the RL agent decides on actions based on information different from the simulation environment. To effectively guide the policy search, the state space should contain sufficient information. Consequently, the size of the state space in complex environments can become huge. With an enormous state space, it

becomes notoriously hard to visit sufficient state–actions pairs for the *Q*-learning algorithm to converge. The second part of the challenge therefore lies in deciding the size of the state space concerning the training duration of the *Q*-learning algorithm. In this section, we focus on developing two state spaces, differing in construction from available observations. Later on, we analyze the influence of the state space on the agent's performance and feasibility.

We specify the state consisting of an internal and an external part $s_t = (s_t^{int}, s_t^{ext})$. The purpose of the internal part is to describe processes inside the pipe. The role of the external part is to guide policy search by including external environment information in the state space. Ideally, we would like to include predictions of heat demand and electricity price from the current time-step *t* to the end of the episode in the external part of the state space. However, because of the limitation on the state space size, we include only one-hour ahead predictions. To leverage future trends in heat demand, we also incorporate the season and time of the day information in the external state space. The external part of the state consists of season, time of the day, heat demand, and electricity price: $s_t^{ext} = (ss_t, td_t, Q_t, c_t)$.

In every time-step *t*, the simulation environment provides observations corresponding to the available real-world sensory output: temperature at the inlet $\tilde{\tau}_{s,t}^{in}$ and outlet of the pipeline $\tilde{\tau}_{s,t}^{out}$, and mass flow $\tilde{m}_{s,t}$. These observations update the internal state. To examine the impact of the form of observations on the performance of the RL agent, we construct two internal state spaces—a full and a partial state. They provide information to the RL agent about its environment with different granularity levels. In Figure 2, the interaction between the environment and the RL agent with two internal state space representations is visualized. The creation of the full and partial state space is presented in the following two subsections.

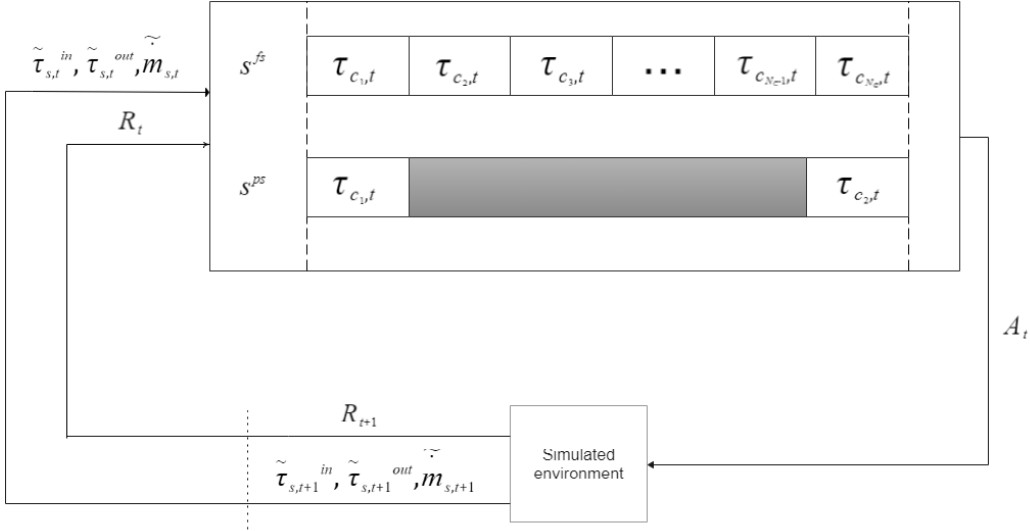

**Figure 2.** The reinforcement learning (RL) agent–simulation environment interaction. Based on the RL agent's action at the time-step *t*, $A_t$, the simulation environment at the time-step $t + 1$ outputs observations of the supply network inlet temperature $\tilde{\tau}_{s,t+1}^{in}$, supply network outlet temperature $\tilde{\tau}_{s,t+1}^{out}$ and mass flow $\tilde{m}_{s,t+1}$, and reward $R_{t+1}$. These observations form a *Q*-learning full $s^{fs}$ and partial state $s^{ps}$.

### 3.2.1. Full State Space

A full state space $Q^{full}$ is implemented as a first-in, first-out queue of water chunks in the supply pipe. The water chunk *i* is characterized by its mass $m_i$ and temperature $\tau_{c_i}$. Including the history of observations forms a Markovian state as the pipe is of finite length *L*, and we can determine when the chunk is pushed out of the pipe [33]. Mass and temperature values are discretized:

$$s_t^{int,f} = (\tau_{c_1,t}, \tau_{c_2,t}, \dots, \tau_{c_{N_C},t}), \quad \tau_{c_1,t} = \tilde{\tau}_{s,t}^{in}, \quad t \in T, \tag{12}$$

where $N_C$ is the number of water chunks:

$$N_C = \lfloor \frac{\rho A_p L}{\Delta m_c} \rfloor + 1, \tag{13}$$

with $\rho$ being the water density, $A_p$ being the surface area of the pipe's cross-section, and $\Delta m_c$ being the mass flow discretization step.

The mass of the water chunk is the product of the mass flow and the optimization interval $\Delta t$:

$$m_{c,t} = \widetilde{\dot{m}}_{s,t} \Delta t, \quad c = 1, \ldots, N_C, \quad t \in T \tag{14}$$

If the mass of the water chunk is larger than the discretization step $m_{c,t} \geqslant \Delta m_c$, the chunk is represented as the $n_c$ number of chunks of the same temperature:

$$n_c = \lfloor \frac{m_{c,t}}{\Delta m_c} \rfloor, \quad n_c \leqslant N_C \tag{15}$$

The temperature of each chunk is rescaled according to Newton's cooling law through subsequent time-steps:

$$\tau_{c,t} = \tau_{env} + (\tau_{c,t-1} - \tau_{env}) \exp^{-\frac{h A_{sur} \Delta t}{C \Delta m_c}}, \quad c = 1, \ldots, N_C, \quad t \in T, \tag{16}$$

where $\tau_{env}$ is the temperature of the surrounding ground, $C$ is the heat capacity of the water, $h$ is the thermal resistance, and $A_{sur}$ is the surface area of the pipe. This representation of heat losses eliminates the need for time-delays and outlet temperature approximation, as explained in Section 2.1, while preserving stability during training guaranteed by the convergence theorem of the tabular $Q$-learning [30].

The size of this state space is the size of the Cartesian product $N_\tau^{N_C} = \underbrace{N_\tau \bigotimes N_\tau \bigotimes \ldots N_\tau}_{N_C}$,

where $N_\tau = \lfloor \frac{\tau_s^{max} - \tau_s^{min}}{\Delta \tau} \rfloor$. The $\Delta \tau$ is the temperature discretization step.

### 3.2.2. Partial State Space

A partial state space $Q^{par}$ is formed directly from the discretized observations, and it consists of the temperature at the inlet and outlet of the supply network pipe and mass flow:

$$s_t^{int,p} = (\tau_{c_1,t}, \tau_{c_2,t}, \dot{m}_{c,t}), \quad \tau_{c_1,t} = \widetilde{\tau}_{s,t}^{in}, \quad \tau_{c_2,t} = \widetilde{\tau}_{s,t}^{out}, \quad \dot{m}_{c,t} = \widetilde{\dot{m}}_{s,t}, \quad t \in T \tag{17}$$

The size of this state space is the size of the Cartesian product $N_C \bigotimes N_\tau \bigotimes N_\tau$, where $N_C = \lfloor \frac{\dot{m}^{max}}{\Delta \dot{m}_c} \rfloor$, and $N_\tau$ is defined as in Section 3.2.1.

### 3.3. Reward Engineering

The agent receives a reward from the environment following from its actions. The reward signal is calculated independently of the agent, and it represents the degree of the agent's success. The agent's final goal is to maximize the cumulative reward collected through an episode [32]. In this paper, the aim is profit maximization while satisfying the heat demand of consumers, and keeping the mass flow, inlet and outlet temperatures inside the feasible region. A common approach in the literature is to incorporate objective functions and safety constraints into the reward function [34]. Therefore, the challenge becomes multi-objective optimization. To achieve the desired behavior of the $Q$-learning algorithm, we carefully determine the shape and hyperparameters' values for composing the sub-reward functions.

A reward signal is the linear sum of sub-rewards concerning profit $R_t^{profit}$, under-delivered heat $R_t^{h,under}$, maximal inlet supply temperature $R_t^{max,s,in}$, minimal inlet supply temperature $R_t^{min,s,in}$, minimal inlet return temperature $R_t^{min,r,in}$, and maximal mass flow $R_t^{max,m}$ functions:

$$R_t = R_t^{profit} + R_t^{h,under} + R_t^{max,s,in} + R_t^{min,s,in} + R_t^{min,r,in} + R_t^{max,m}, \quad t \in T \tag{18}$$

To achieve the desired trade-off between profit maximization and constraints satisfaction, we experimentally determine the hyperparameters' values of sub-rewards. According to these values, each sub-reward signal is rescaled. For accelerating the learning process of the algorithms, the sub-reward signal forms a symmetric function with a sharp gradient toward maximal value. In the following sections, the formation of sub-rewards is explained.

### 3.3.1. Profit Sub-Reward

A rescaling of the profit function in Equation (3) requires determining an upper and a lower bound on the function.

We can identify three cases for the upper bound $R^{profit,max}$ depending on the placement of the maximal electricity price $c^{max}$ between cost coefficients of heat $a_0$ and electricity production $a_1$. If the maximal electricity price is greater than the price of the electricity production, the maximum profit is achieved in the extreme point $(H_3, P_3)$ of the CHP unit operating region. If the maximal electricity price is placed between $-2 \cdot a_0 + a_1$ and $a_1$, the maximum profit is achieved in the point $(H_0, P_0)$. Otherwise, the maximum profit is achieved in the point $(H_1, P_1)$:

$$R^{profit,max} = \begin{cases} -a_0 \cdot H_3 + (c^{max} - a_1) \cdot P_3, & \text{if} \quad c^{max} > a_1 \\ -a_0 \cdot H_0 + (c^{max} - a_1) \cdot P_0, & \text{if} \quad -2 \cdot a_0 + a_1 < c^{max} \leqslant a_1 \\ -a_0 \cdot H_1 + (c^{max} - a_1) \cdot P_1, & \text{otherwise} \end{cases} \tag{19}$$

Depending on the value of the minimal electricity price $c^{min}$, we distinguish three cases for the lower bound $R^{profit,min}$. If the minimal electricity price is less than zero, the minimal profit is in the extreme point $(H_3, P_3)$. If the minimal electricity price is placed between zero and $2 \cdot a_0 + a_1$, the minimal profit is achieved in the point $(H_2, P_2)$. Otherwise, the minimal profit is in the point $(H_1, P_1)$:

$$R^{profit,min} = \begin{cases} -a_0 \cdot H_3 + (c^{min} - a_1) \cdot P_3, & \text{if} \quad c^{min} < 0 \\ -a_0 \cdot H_2 + (c^{min} - a_1) \cdot P_2, & \text{if} \quad 0 \leqslant c^{min} < 2 \cdot a_0 + a_1 \\ -a_0 \cdot H_1 + (c^{min} - a_1) \cdot P_1, & \text{otherwise} \end{cases} \tag{20}$$

The profit function $F_t$ is rescaled to fit an artificial upper $\overline{R}^{profit}$ and lower bound $\underline{R}^{profit}$ with a function gradient $\xi_p$:

$$R_t^{profit} = \overline{R}^{profit} - (\overline{R}^{profit} - \underline{R}^{profit}) \left( \frac{|F_t - R^{profit,max}|}{|R^{profit,max} - R^{profit,min}|} \right)^{\xi_p}, \quad t \in T \tag{21}$$

### 3.3.2. Underdelivered Heat Sub-Reward

One of the requirements is to satisfy the consumer's heat demand, which is represented by Equation (6). This requirement is also supported by constraints on the minimum supply and return inlet temperature by Equations (5) and (7). The heat is underdelivered when the actual delivered heat to the consumer measured by the simulator $\tilde{h}_t^c$ is less than the consumer's heat demand $Q_t$. The largest mismatch between the delivered heat and heat demand occurs when the heat delivered to the consumer is minimal $h^{c,min} = 0$ and the heat demand is maximal $Q^{max}$. The penalty for underdelivered heat is rescaled function with hyperparameters, an upper $\overline{R}^{h,under}$ and lower bound $\underline{R}^{h,under}$, and gradient $\xi_h$:

$$R_t^{h,under} = \overline{R}^{h,under} - (\overline{R}^{h,under} - \underline{R}^{h,under}) \left( \frac{|\tilde{h}_t^c - Q_t|}{|h^{c,min} - Q^{max}|} \right)^{\xi_h}, \quad t \in T \tag{22}$$

### 3.3.3. Minimal and Maximal Temperature Sub-Reward

The negative reward is given if the temperature values in the supply and/or the return network violate the upper or lower bound specified by Equations (5) and (7). The largest possible violation is $\Delta\tau^{max} = \tau_s^{max} - \tau_r^{min}$. The smallest violation is $\Delta\tau^{min} = 0$. The reward function is rescaled with hyperparameters, upper $\overline{R}^\tau$ and lower bound $\underline{R}^\tau$, and gradient $\xi_\tau$.

The penalty for the supply network inlet temperature measured by the simulator $\widetilde{\tau}_{s,t}^{in}$ being higher than the upper bound $\tau_s^{max}$ is:

$$R_t^{max,s,in} = \overline{R}^\tau - (\overline{R}^\tau - \underline{R}^\tau)\left(\frac{\widetilde{\tau}_{s,t}^{in} - \tau_s^{max} - \Delta\tau^{min}}{\Delta\tau^{max} - \Delta\tau^{min}}\right)^{\xi_\tau}, \quad t \in T \tag{23}$$

The penalty for the supply network inlet temperature being lower than the bound $\tau_s^{min}$ is:

$$R_t^{min,s,in} = \overline{R}^\tau - (\overline{R}^\tau - \underline{R}^\tau)\left(\frac{\tau_s^{min} - \widetilde{\tau}_{s,t}^{in} - \Delta\tau^{min}}{\Delta\tau^{max} - \Delta\tau^{min}}\right)^{\xi_\tau}, \quad t \in T \tag{24}$$

If the return network inlet temperature $\widetilde{\tau}_{r,t}^{in}$ is lower than the bound $\tau_r^{hes,min}$, the punishment is:

$$R_t^{min,r,in} = \overline{R}^\tau - (\overline{R}^\tau - \underline{R}^\tau)\left(\frac{\tau_r^{hes,min} - \widetilde{\tau}_{r,t}^{in} - \Delta\tau^{min}}{\Delta\tau^{max} - \Delta\tau^{min}}\right)^{\xi_\tau}, \quad t \in T \tag{25}$$

### 3.3.4. Maximal Mass Flow Sub-Reward

The requirement that the mass flow is lower than the upper bound is specified in Equation (8). This requirement is translated into the sub-reward function:

$$R_t^{max,m} = \overline{R}^m - (\overline{R}^m - \underline{R}^m)\left(\frac{\dot{m}_{c,t} - \dot{m}^{max} - \Delta\dot{m}^{min}}{\Delta\dot{m}^{max} - \Delta\dot{m}^{min}}\right)^{\xi_m}, \quad t \in T \tag{26}$$

The hyperparameters in the reward function are chosen to encourage learning of the desired behavior: profit gain and satisfaction of constraint requirements. We explain the choice of these parameters in the next section.

## 4. Experimental Design

To understand the strengths and weaknesses of MINLP and the *Q*-learning algorithm, in particular in dealing with the environment and model uncertainty under limited sensor information, we evaluate them on two grid settings and two electricity markets. The grid structure and parameter values are chosen to enable a fair comparison of two algorithms and realistic grid limitations. Their values and the rationale behind the choices are presented in this section.

The CHP economic dispatch model (Section 2.1), even for a simple grid with one producer and one consumer, is highly complex. Scaling and moderating the degree of nonlinearity is essential for the stability of mathematical optimization [35]. Increasing the number of consumers leads to an increase in the DHN model complexity and a decrease in the numerical stability of the MINLP. Therefore, to fairly evaluate the two algorithms (*Q*-learning and MINLP), the DHN in our experiments consists of one CHP production plant and one consumer, which are directly connected by a supply and a return pipeline. The pipe's diameter (both in the supply and return network) is 0.5958 m. We consider a network with a short pipeline $L = 4$ km and with a longer pipeline $L = 12$ km. There are two reasons for choosing these lengths. The first one is to evaluate whether pipeline energy storage and consequently profit gain increases with an increase of stored water capacity. The second is to inspect how an increase in environment stochasticity, the consequence

of the pipe's length increase, influences the performance and feasibility properties of the *Q*-learning algorithm.

In Table 1, the DHN parameter values used in the experiments are listed. The base CHP extreme points $(H_i, P_i)$ for $i \in \{1, 2, 3, 4\}$ and the fuel price ($a_0$ for the heat and $a_1$ for the electricity) are taken from [22]. These CHP extreme points, and the heat and electricity production price coefficients, are used in the optimizers and the simulator according to Bloess [36]. The minimal heat load pressure at the HES $pr^{min}$ is specified according to Euroheat guidelines [37]. Minimal and maximal supply inlet temperature ($\tau_s^{min}, \tau_s^{max}$), minimal return inlet temperature $\tau_r^{hes,min}$, and maximal mass flow $\dot{m}^{max}$ limits are determined in consultation with the collaborator company Flex Technologies, corresponding to realistic grid constraints.

**Table 1.** The district heating network parameters.

| $(H_i, P_i)$ [MW] | $a_0$[e/h] | $a_1$[e/h] | $\tau_s^{min}$[°C] | $\tau_s^{max}$[°C] | $\dot{m}^{max}[\frac{m}{s}]$ | $\tau_r^{hes,min}$[°C] | $pr^{min}$[kPa] |
|---|---|---|---|---|---|---|---|
| (0,10), (10,5), (70,35), (0,50) | 8.1817 | 38.1805 | 70 | 110 | 3 | 45 | 100 |

The electricity price data were downloaded from the ENTSOE website [38], and heat demand data were estimated by Ruhnau et al. [39] according to the German gas standard load profile approach defined by the German Association of Energy and Water Industries. Both datasets consist of hourly values over a five-year period for The Netherlands. Therefore, the optimization interval is one hour. The heat demand data are rescaled so that the maximum corresponds to the maximal heat production of the CHP plant. The data are discretized to enable integration of heat demand and electricity price in the external state space $s_t^{ext}$ of the *Q*-learning algorithm. Corresponding to the length of a day, we choose the length of an episode to be 24 h.

To inspect the adaptability and time-scale flexibility properties of MINLP and the *Q*-learning algorithm, we apply them to trading on two electricity markets. These two types of markets are deterministic and uncertain electricity markets, which are called day-ahead and intraday electricity markets.

The CHP economic dispatch model for one day is solved in ten iterations, as in [12]. The iteration length is six minutes. These values are chosen to strengthen convergence guarantees of the optimization procedure and to eliminate insufficient optimization intervals as the cause of instability or non-convergence. An interior-point optimizer (IPOPT) [40] is used for solving nonlinear subproblems.

To evaluate the performance of *Q*-learning on unseen scenarios of the testing dataset, five years of data are split into a training and a testing dataset. Before the split, to enable the diversity of the training data, the dataset is randomly shuffled on a day-to-day (an episode) basis. A training–testing ratio is set to 0.9 to exploit more training data. The testing dataset consists of 182 days. The MINLP uses data from the testing dataset. The state space size is 2,250,000, and the Q-matrix size is 38,250,000 for both $Q^{full}$ and $Q^{par}$. The number of training episodes of *Q*-learning is 25,000,000 and is chosen as the compromise between the time limit for the algorithm's training and visiting sufficient state–action pairs.

The reward signal consists of the linear sum of six sub-rewards as defined by Equation (18). Constraints on temperature and mass flow are considered hard constraints. Their violation leads to the termination of the episode. Therefore, the upper bound of the penalty for temperature and mass flow violation ($\overline{R}^\tau, \overline{R}^m$) is equal to the worst possible scenario (lower bound) of profit $\underline{R}^{profit}$ and underdelivered heat $\underline{R}^{h,under}$ during the entire episode (24 h). Terminating the episode reduces the state space size for *Q*-learning. The penalty value is chosen to prevent reaching a terminal state as quickly as possible to avoid accumulating penalties. In Table 2, hyperparameters of rescaled sub-reward functions are listed.

**Table 2.** The reward function parameters.

| $\overline{R}^{profit}$ | $\underline{R}^{profit}$ | $\overline{R}^{h,under}$ | $\underline{R}^{h,under}$ | $\overline{R}^{\tau},\overline{R}^{m}$ | $\underline{R}^{\tau},\underline{R}^{m}$ |
|---|---|---|---|---|---|
| 20 | −10 | 0 | −40 | $24 \times (-10 - 40)$ | $2 \times 24 \times (-10 - 40)$ |

Temperature and mass flow discretization errors cause the environment to behave stochastically. Therefore, the learning rate $\alpha = 0.8$ and exploration–exploitation parameter $\epsilon = 0.8$ are set to high values at the beginning of the training process. The violation of mass flow and temperature constraints has a long-term propagating effect on the reward of the episode. Consequently, the discount factor $\gamma = 0.95$ is set to a high value. The Q-matrix initialization $Q(s_0, a_0)$ influences the exploration–exploitation ratio [41]. A Q-matrix initialized with negative values has lower exploration and finds a safer policy (which does not violate temperature and mass flow constraints) more quickly. This might be beneficial for larger networks when the stochasticity of the environment, and therefore a chance for choosing unsafe actions, is higher.

## 5. Overview of the Simulator

In previous sections, we describe two algorithms for the CHP economic dispatch with grid dynamics—MINLP and Q-learning algorithm, and their configuration. These algorithms are assessed on four criteria: performance, stability, feasibility, and time-flexibility. Ideally, we would evaluate the performance and feasibility of these algorithms on the real-world grid. An aggravating factor for this type of evaluation is a high safety risk. The majority of previous works evaluate the algorithm's outcome on the same model or do not evaluate it at all. To enable a fair and objective comparison of algorithms in an environment resembling the real world, we developed a detailed simulation environment of processes in the district heating network. In this section, we briefly describe this simulator of a district heating grid and explain how we measure the criteria using this simulator. The simulator is owned by the company Flex Technologies, and the writing of a detailed technical report is in progress [42].

The input to the simulator is the heat and electricity production $(h_t, p_t)$ determined by the optimization algorithm. Other inputs are the consumer's heat demand $Q_t$ and electricity price $c_t$. The simulator shares the grid setting, physics constants, and parameter values with the mathematical optimizer and Q-learning algorithm, as listed in Section 4. Firstly, the grid object consisting of a CHP unit as a producer, supply edge, return edge and HES is created. Each edge consists of water plugs characterized by the mass and temperature. A secondary-side mass flow at the time-step $t$, $\widetilde{m}_{s,t}^{sec}$, is calculated based on the known heat demand and the difference between initial secondary-side supply inlet $\widetilde{\tau}_{s,t}^{in,sec}$ and secondary-side return outlet temperature $\widetilde{\tau}_{r,t}^{out,sec}$:

$$\widetilde{m}_{s,t}^{sec} = \frac{Q_t}{\widetilde{\tau}_{s,0}^{in,sec} - \widetilde{\tau}_{r,0}^{out,sec}} \tag{27}$$

Initial secondary-side temperatures are assumed constant $\widetilde{\tau}_{s,0}^{in} = 70\,^{\circ}\text{C}$ and $\widetilde{\tau}_{r,0}^{out,sec} = 45\,^{\circ}\text{C}$. The primary-side mass flow $\widetilde{m}_{s,t}$, primary-side return inlet temperature $\widetilde{\tau}_{r,t}^{in}$, and secondary-side supply inlet temperature $\widetilde{\tau}_{s,t}^{in,sec}$ are determined by the heat exchange station. The heat exchange process is developed based on the logarithmic mean temperature difference method [43]. The return network outlet temperature $\widetilde{\tau}_{r,t}^{out}$ is calculated by pushing out plugs of the return pipe conditioned on $\widetilde{m}_{s,t}$ and $\widetilde{\tau}_{r,t}^{in}$. Following Equation (4), and known $h_t$, $\widetilde{m}_{s,t}$ and $\widetilde{\tau}_{r,t}^{out}$, the primary-side supply inlet temperature $\widetilde{\tau}_{s,t}^{in}$ is computed. By pushing out plugs from the supply pipe based on $\widetilde{\tau}_{s,t}^{in}$ and $\widetilde{m}_{s,t}$, the outlet temperature of the supply network $\widetilde{\tau}_{s,t}^{out}$ is calculated. Finally, the simulator computes the actual delivered heat to the consumer $\widetilde{h}_t^c$ using pre-calculated $\widetilde{m}_{s,t}$, $\widetilde{\tau}_{s,t}^{out}$ and $\widetilde{\tau}_{r,t}^{in}$ in Equation (6). The model of the CHP

unit in the simulator is the same as in the mathematical optimizer. Therefore, the profit is calculated by Equation (3).

The performance and feasibility are measured by the simulator's output—profit gain (in €) and percentage of violations. Violations are assessed on five points: under-delivered heat with respect to consumer's heat demand $\tilde{h}_t^c < Q_t$, supply inlet temperature higher than the maximum $\tilde{\tau}_{s,t}^{in} > \tau_s^{max}$, supply inlet temperature lower than the minimum $\tilde{\tau}_{s,t}^{in} < \tau_s^{min}$, return inlet temperature lower than the minimum $\tilde{\tau}_{r,t}^{in} < \tau_r^{hes,min}$, and mass flow higher than the upper bound $\tilde{\dot{m}}_{s,t} > \dot{m}^{max}$. Values of minimum and maximum parameters are listed in Table 1.

## 6. Experimental Results on Two Case Studies

The main purpose of our study is to understand the properties of two proposed algorithmic approaches in exploiting the flexibility of pipeline energy storage, robustness for days with varying heat demand and electricity prices, safety, and operation in deterministic and uncertain environments. Firstly, we analyze the differences between using a full $Q^{full}$ and a partial $Q^{par}$ state space model for $Q$-learning. Then, to understand the potential of $Q$-learning, we benchmark the best-performing one against the state-of-the-art nonlinear programming approach (MINLP). The source code of all algorithms for the case studies is available at [44].

### 6.1. Full versus Partial State Space Q-Learning

To analyze how the learning process and performance differ for the full and partial state space model for $L = 4$ km and $L = 12$ km, we plot the cumulative reward function and updates to two $Q$-value functions during the training stage (Figure 3). To enable diversity in analyzed $Q$-value functions, we randomly choose two $Q$-values that contain the summer and winter season parts of the state space.

The reward function and two $Q$-value functions during the training stage for the partial state are almost constant. That means the agent does not learn the system's dynamics from interactions with the environment when the internal state space consists only of available sensory inputs. The reason is that temperatures at the pipe's inlet and outlet and mass flow at the time-step $t$ can correspond to many underlying true states. The control strategy is incapable of inferring underlying states with the trial-and-error approach. Consequently, $Q^{par}$ violates the safety constraints of the temperature and mass flow, which are part of a reward function (Section 3.3) in every episode of the testing dataset, leading to premature episode ending. As available sensor measurements are insufficient for control and safety, $Q^{par}$ is excluded from the further comparison.

### 6.2. Full State Space Q-learning versus Mixed-Integer Nonlinear Program

Here, we compare $Q^{full}$ and MINLP on the day-ahead and real-time electricity market on four criteria: performance, stability, feasibility, and time flexibility to understand profit potential, robustness, feasibility limitations, and applicability to specific scenarios of each approach. We elaborate on experimental results from the algorithmic and operation of district heating system sides and analyze how different criteria influence each other.

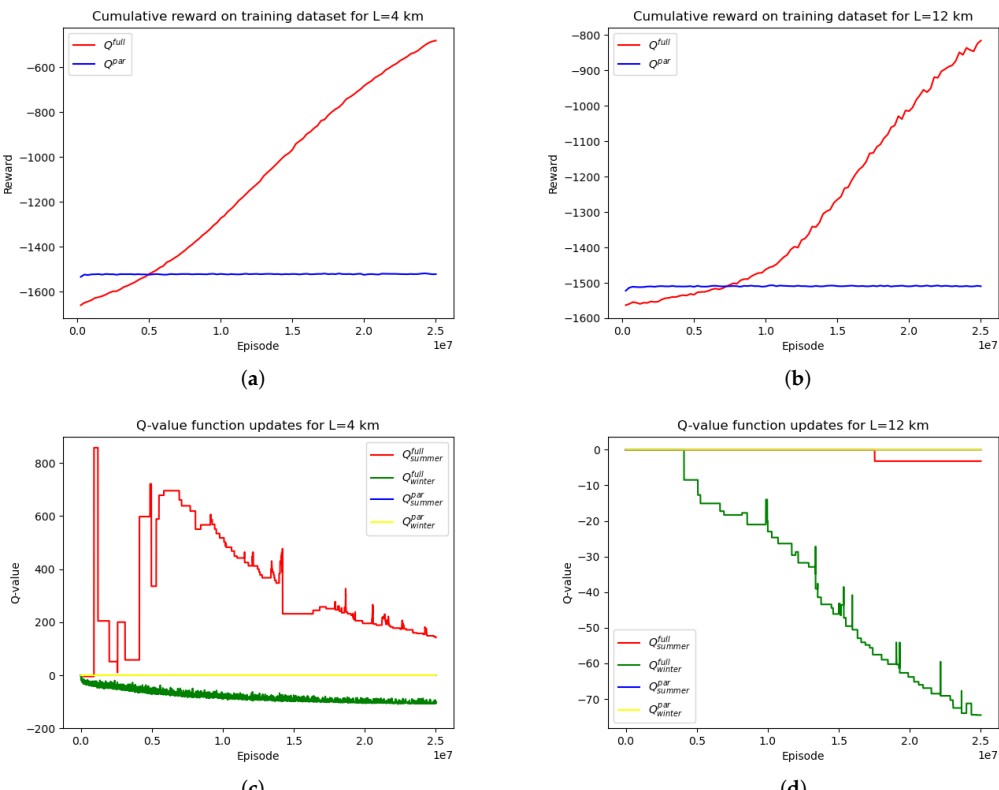

**Figure 3.** (**a**) Cumulative reward change during training for $L = 4$ km. (**b**) Cumulative reward change during training for $L = 12$ km. (**c**) Q-value functions $Q(s_1, a_1)$ and $Q(s_2, a_2)$ updates for $L = 4$ km. (**d**) Q-value functions $Q(s_1, a_1)$ and $Q(s_2, a_2)$ updates for $L = 12$ km. Q-value functions for $Q^{par}_{summer}$ (blue line) and $Q^{par}_{winter}$ (yellow line) are overlapping.

### 6.2.1. Performance

The cumulative profit over a period of 24 h (an episode) from exploiting the flexibility of pipeline energy storage is visualized on the left side in Figure 4. This is an example day from the testing dataset, for which both MINLP and $Q^{full}$ outputs are feasible.

The performance of MINLP (yellow line) and $Q^{full}$ (green line) is compared to the LP for the CHP economic dispatch without grid dynamics (blue line), as well as the upper bound (red line) and basic control strategy (black line). The LP always finds an optimal solution to the CHP economic dispatch model. Therefore, the profit gain achieved by incorporating the pipeline energy storage can be inspected as the profit differences to the results without modeling storage in the network. These are shown in the left-hand side plots of Figure 4.

In the first nine hours of the day for $L = 4$ km (left top plot in Figure 4), the electricity price is low, and both MINLP and $Q^{full}$ make use of it by producing more heat and less electricity. The electricity price increases over the following time-steps, and MINLP and $Q^{full}$ use part of the already produced heat to satisfy consumer's heat demand. This facilitates an increase in the production of electricity, and surpassing of the LP benchmark, for MINLP in the 12th hour and for $Q^{full}$ in the 14th hour. MINLP enables savings of €2159.41 for $L = 4$ km compared to the LP. The increase in the pipe length to $L = 12$ km leads to an increase in the volume of possible stored water and consequently to a greater opportunity for utilizing the thermal inertia property of the network. Consequently, MINLP for the larger pipe length $L = 12$ km saves €3271.02 compared to the LP. $Q^{full}$ for cases $L = 4$ km and $L = 12$ km saves €6.32 and €115.79, respectively.

The box plots on the right side in Figure 4 summarize results for all days of the testing dataset. The median and quartile values show the lower performance of $Q^{full}$ and the decrease of $Q^{full}$ performance when the pipe's length is increased from $L = 4$ km to

$L = 12$ km on the entire dataset. The differences in the means of achieved profit for LP and MINLP, and LP and $Q^{full}$ are statistically significant with paired *t*-test *p*-value $p < 1\%$. We distinguish three possible explanations for described behavior of $Q^{full}$.

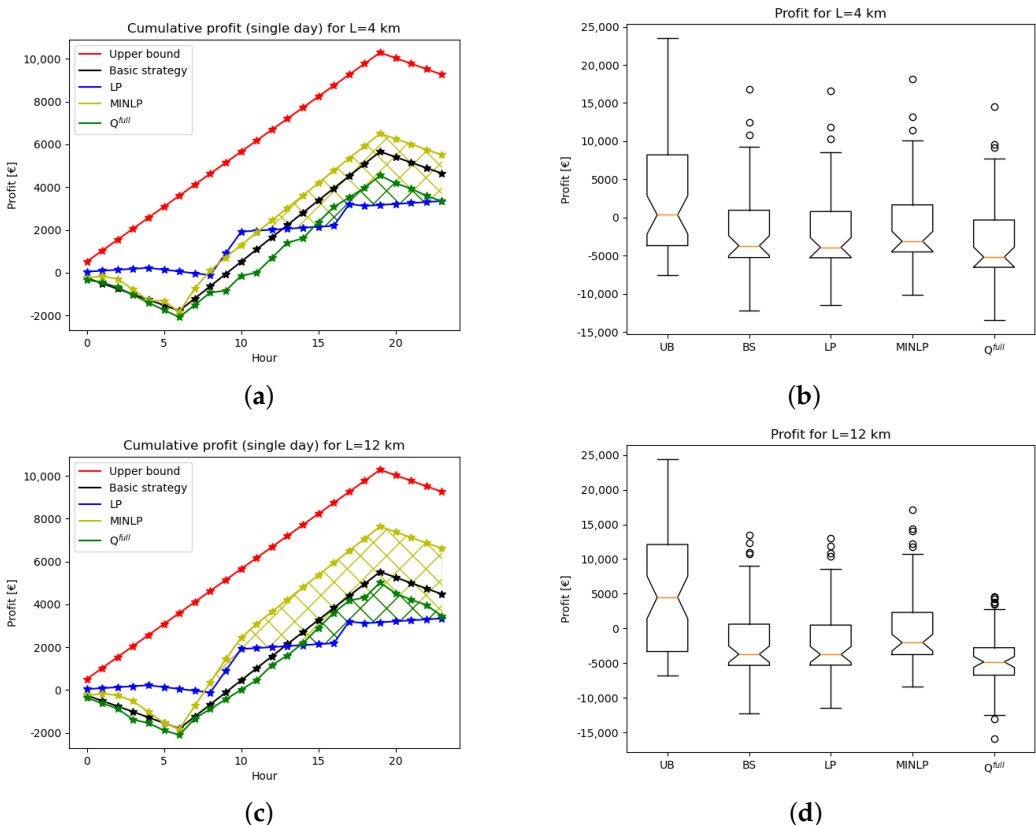

**Figure 4.** (**a**) The single-day cumulative profit for $L = 4$ km. (**b**) The box-plot profit for $L = 4$ km. (**c**) The single-day cumulative profit for $L = 12$ km. (**d**) The box-plot profit for $L = 12$ km.

The first comes from the discretization of the state and action space in tabular *Q*-learning. This discretization introduces an observability error—the nonlinear combination of discretization errors of the temperature in the supply and return network, and the mass flow. The observability error presents the difference between the information about the pipe state available to a *Q*-learning algorithm and the simulator. This error brings the randomness in the environmental information (environment stochasticity) available to the *Q*-learning agent and the actual values in the simulator. A two sample *t*-test with $p < 1\%$ proves the causal effect of observability error on the reward function during testing (when there is no exploration). With enlarging the pipe from 4 to 12 km, the discretization step of chunk's mass $\Delta \dot{m}_c$ is increased three times to keep the state space the same (feasible) size. In the reward function (Section 3.3), the punishment for violating one of the four feasibility points is greater than the worst-case profit collected through an episode. Therefore, the increased environment stochasticity leads to less exploration (Figure 3) and choosing "safer" actions. This results in the performance degradation of $Q^{full}$ when enlarging the pipe (Figure 4).

The second explanation for the profit difference between MINLP and $Q^{full}$ lies in the time horizon of the predictions of heat demand and electricity price available to two algorithms. When creating the mathematical model of the DHN, the MINLP is provided with deterministic (perfect) knowledge of the heat demand and electricity prices 24 h (length of the optimization horizon) in advance. In the $Q^{full}$ algorithm, the information about the environment is part of the state space. Because of the state space size limitation, $Q^{full}$ includes perfect prediction of the heat demand and electricity price only *one* hour in advance through an external part of the state space $s^{ext}$. Additional information to the agent

about heat demand trends is provided through an inclusion of one step ahead season and time of the day information in the external state. The MINLP can be seen as an optimization strategy on the day-ahead electricity market, whereas $Q^{full}$ delivers a trading strategy for the real-time electricity market. The suitability of the $Q$-learning algorithm for online use is important. In the future, real-time services will gain more significance as the share of intermittent and distributed energy sources is increasing [2].

Both MINLP and $Q^{full}$ have limitations when applied to the uncertain and certain electricity markets, respectively. With an inclusion of future electricity prices, the state space of $Q$-learning grows exponentially, making it an unsuitable choice for trading on the day-ahead electricity market. The MINLP can be adjusted to operation on the real-time electricity market with a rolling horizon approach. This approach implies optimization with a 24 h time horizon, application of the first action to the simulator model, initialization of the mathematical model variables with values from the simulator after the transition, and repetition of the optimization. Due to complex, nonlinear dependencies, the model is sensitive to external input from the simulator used for the initialization. The interaction with the simulator results in an instability of the optimization (more details on the stability of the optimization in Section 6.2.2). When applying the rolling horizon approach to all the days in the testing dataset, the furthest reached time-step in the episode is 22 (out of 24): no single day in the data set could be completed.

To gain additional insight into the influence of the environment stochasticity and the horizon of perfect information of heat demand and electricity price on the performance of the algorithms, we plot the profit dependence on variance. The heat demand variance does not have a distinguished influence pattern on the algorithms' performance (left side in Figure 5). An increase in the electricity price variance increases profit variance (right side in Figure 5). The performance difference between MINLP and $Q^{full}$ is remarkable when analyzed for the electricity price variance and the long pipe length of $L = 12$ km (bottom-right plot in Figure 5). We identify two possible reasons for the differences between the upper and lower quartile and the maximum and minimum of box plots of profit in Figure 5. The first is that the heat demand and/or electricity price for different days can have the same variance, but the pattern change from time-step to time-step can vary. Specific scenarios (for example, lower heat demand in a few time steps followed by higher heat demand) facilitate the use of pipeline energy storage flexibility; therefore, the profit will be in the upper quartile of the box plot for those scenarios. The second reason is that heat demand and electricity price impacts on profit gain are intercoupled. Consequently, the achieved profit on days where heat demand has the same variance can vary depending on the electricity price pattern.

The third explanation for the profit difference between MINLP and $Q^{full}$ is the behavior of these algorithms at the end of the optimization time horizon. At the end of the time horizon, the algorithm can exploit the residual heat in the pipeline to maximize the profit gain. This phenomenon we call "draining the pipe". As the CHP economic dispatch is a continuous process, draining the pipe is undesirable. A $Q$-learning algorithm assigns values to state–action pairs through interactions with the environment. The state–action pair at the end of an episode might be at the beginning of the following episode. Therefore, a $Q$-learning algorithm does not learn to "drain the pipe", since it is updating the same $Q$-values in different time steps. The chance to exploit the heat residual at the end of the episode gives the advantage to MINLP in achieving profit.

### 6.2.2. Stability

As explained in Section 2.1, the model of CHP economic dispatch is discontinuous, nonconvex, and highly nonlinear because of dependencies of temperature at the inlet and outlet of the pipeline in the supply and return network and mass flows. The complexity of the model represents a challenge for optimization with any state-of-the-art solver.

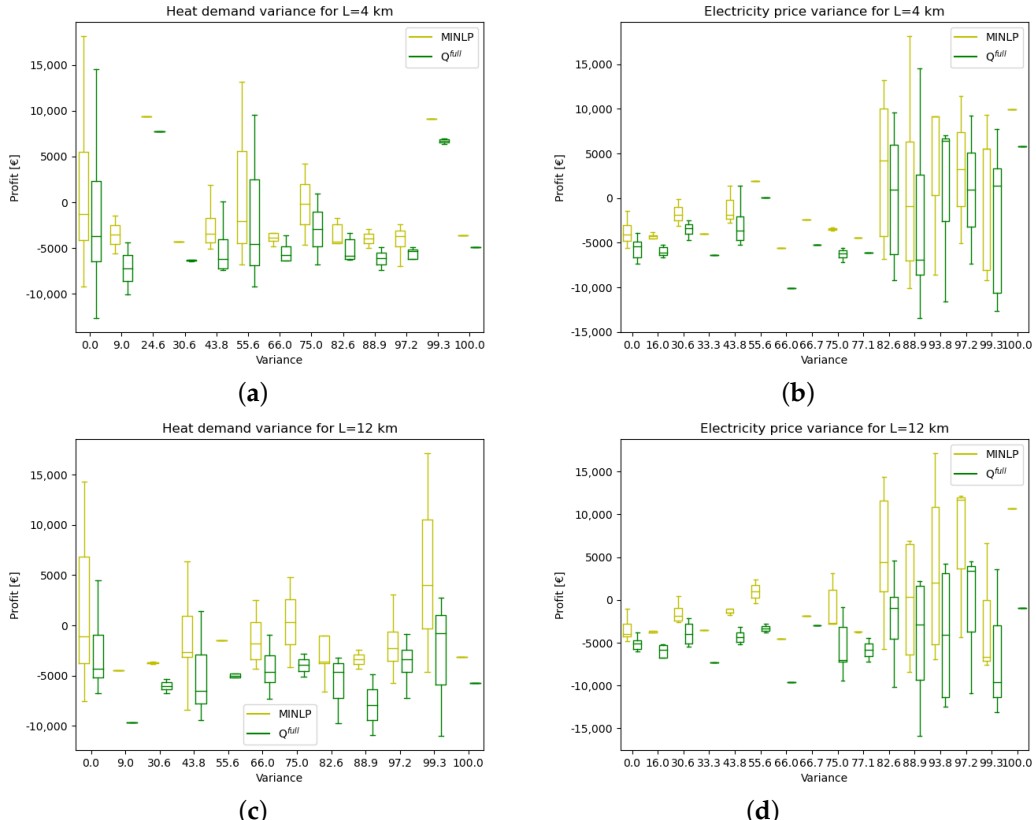

**Figure 5.** (**a**) Heat demand variance for $L = 4$ km. (**b**) Electricity price variance for $L = 4$ km. (**c**) Heat demand variance for $L = 12$ km. (**d**) Electricity price variance for $L = 12$ km.

The number of variables in the mathematical model is 553, and the model has 865 constraints. IPOPT of the SCIP optimization suite [45] is used for solving the CHP economic dispatch model on 182 days of the test dataset. With ten iterations, each lasting six minutes, IPOPT does not find a primal bound of the model for 120 days for pipe length $L = 4$ km and 113 days for pipe length $L = 12$ km (Figure 6). These days are omitted from the profit and feasibility analysis for both MINLP and $Q^{full}$ (although $Q^{full}$ finds the solution for all days of the test dataset).

Moreover, the optimization procedure is sensitive to changes in the parameters. The change of parameter $\tau_s^{max}$ from 110 to 120 °C results in an increase of stable days, 92 for $L = 4$ km and 101 for $L = 12$ km. The increase in the range of allowed temperature values relaxes constraints and enables for more solutions to be found.

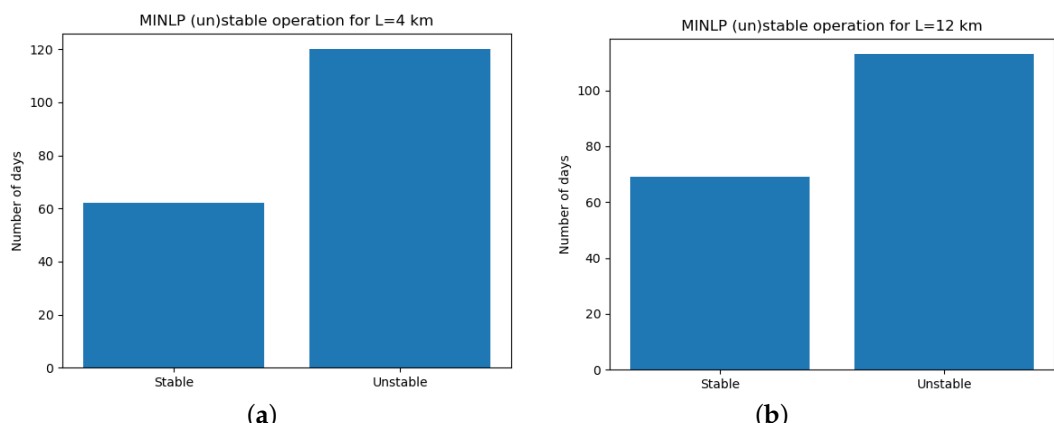

**Figure 6.** (**a**) Stable days for $L = 4$ km. (**b**) Stable days for $L = 12$ km.

### 6.2.3. Feasibility

The feasibility of the CHP economic dispatch model and $Q^{full}$ is accessed on five points by simulator evaluation: underdelivered heat to the consumer, maximum inlet supply temperature, minimum inlet supply temperature, minimum inlet return temperature, and maximum mass flow. In Figure 7, the so-called quantile plots show the percentage of days for which a lower or equal percentage of violation on the vertical axis occurred. No violations of the maximum mass flow and minimum return inlet temperature take place.

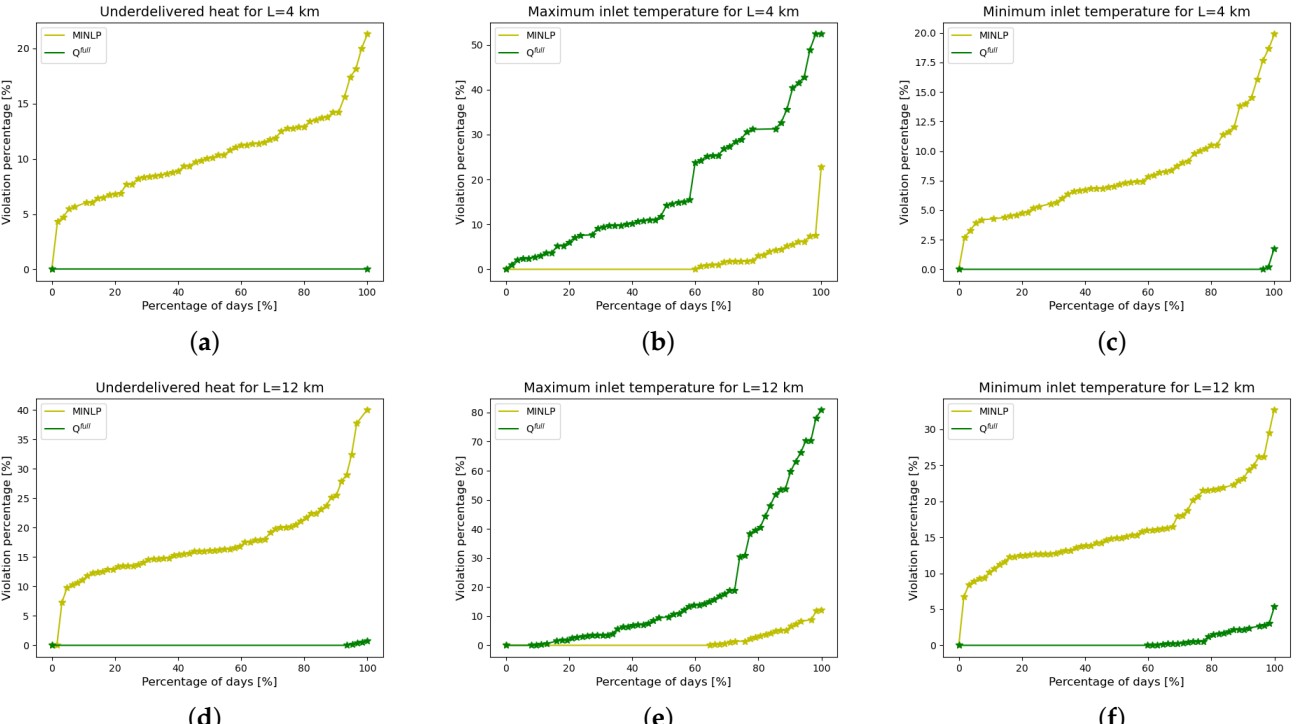

**Figure 7.** (**a**) Underdelivered heat violation for $L = 4$ km. (**b**) Maximum inlet supply temperature violation for $L = 4$ km. (**c**) Minimum inlet supply temperature violation for $L = 4$ km. (**d**) Underdelivered heat violation for $L = 12$ km. (**e**) Maximum inlet supply temperature violation for $L = 12$ km. (**f**) Minimum inlet supply temperature violation for $L = 12$ km.

In the CHP economic dispatch model, network safety guarantees and limitations are implemented as hard constraints, while in the $Q^{full}$ algorithm, they are soft constraints integrated as part of the reward function. Therefore, we expected that the MINLP has a lower percentage of violations, especially related to the underdelivered heat demand. However, this is not the case.

While Li et al. [12] provides a comprehensive mathematical formulation of processes in the DHN pipeline, they assume a constant mass flow between the supply and return network. That is unfavorable in the DHN operation as it can result in high return temperatures [46]. The heat exchange process in the HES is described accurately only with a set of complex mathematical equations. We hypothesize that the simplified model of HES in MINLP causes the divergence between the solution of MINLP and the simulator. To evaluate this hypothesis, we designed two more experiments.

In the first experiment, the simulator uses the HES that controls mass flow (realistic HES). In the second experiment, the simulator uses a simple HES model (ideal HES). The divergence is judged by the mean difference of return outlet temperature between the CHP economic dispatch model and the simulator for both cases. The return outlet temperature is chosen because it is at the end of the temperature propagation cycle, and the difference between the simulator and optimizer should be the most noticeable there. The mean temperature difference between the simulator with the realistic HES and the optimizer is 9 °C (higher temperature in the optimizer), while it is 0.18 °C between the simulator

with the ideal HES and the optimizer. The conclusion is that the absence of a more realistic HES in the CHP economic dispatch model causes the divergence between the solution of MINLP and the simulator.

By interacting with the environment, the $Q^{full}$ algorithm approximates dynamics of the HES, leading to fewer violations. The pipe length increase deepens the environment stochasticity, as explained in Section 6.2.1, resulting in an increase of violations when transferring from pipe length $L = 4$ km to $L = 12$ km.

### 6.2.4. Time-Scale Flexibility

To determine the suitability of algorithms for online use, we access the time flexibility property of the MINLP and $Q^{full}$ algorithm.

The maximum number of iterations for MINLP is ten, and the iteration length is six minutes. The optimization time for one day from the testing dataset is one hour.

The training times of the $Q^{full}$ algorithm for pipe lengths $L = 4$ km and $L = 12$ km are 4744 and 4222 min, respectively. All the experiments are performed on a computer with 4-core Intel I7 8665 CPU. The training times are approximately the same because of the identical state space size. The response on unseen scenarios from the testing dataset by $Q^{full}$ is provided in a few seconds.

Therefore, $Q^{full}$ is the suitable choice for real-time energy trading and encapsulation: the user only needs to input the operating state to get the control strategy, while the optimization algorithm needs to re-write the constraints and other formulas for different situations and repeat optimization.

### 7. Conclusions

The main challenges in automating the control of a district heating system are the uncertainty coming from the environment and from the model inaccuracies under limited sensory information. These difficulties characterize the control of many real-world industrial systems. Exact mathematical modeling and optimization fails to overcome the above-mentioned challenges, as the resulting models are too complex.

To address such operational obstacles in the DHN control, we proposed and analyzed a *Q*-learning algorithm for the CHP economic dispatch utilizing pipeline energy storage. We modeled the district heating system as an MDP with full and partial state space from the representative sensory information. The reward signal was designed to lead the agent to the goal and set the constraints on the system's limitations.

Our newly proposed approach based on *Q*-learning with a full state space model successfully approximates the complex dynamics of a DHN through interactions with a simulation environment. Consequently, when compared with a nonlinear optimization method by simulator evaluation, it results in fewer constraint violations. The mathematical optimizer does not find a primal bound of the model for 120 days for pipe length $L = 4$ km and 113 days for pipe length $L = 12$ km. In contrast to the mathematical optimizer, the *Q*-learning algorithm provides a stable output for all evaluation days. The obtained results show that *Q*-learning requires more computational power during algorithm training but has better time scale flexibility compared to MINLP (provides the response on unseen scenarios in a few seconds). Therefore, it is more suitable for real-time energy markets.

However, a *Q*-learning algorithm achieves a lower profit gain of order of magnitude €$10^2$ compared to solutions provided by the MINLP of order of magnitude €$10^3$. The algorithm is also sample-inefficient, requiring a long training time. The reward function reflects all optimization goals and safety requirements, making it hard to establish robust safety assurance.

The scalability of our approach is defined by the trade-off between the size of the DHN, the size of the state space, and the training time of a *Q*-learning algorithm. If we want to accurately approximate the environment's dynamics, the size of a *Q*-learning state space will exponentially grow when enlarging the DHN, leading to prolonged training

time. To mitigate this, we can keep the size of the state space constant, but this results in an increase in environment stochasticity, leading to an increase in the violation of constraints.

The heat transfer process in the district heating pipeline has strong spatial–temporal dependence. To exploit this dependence and reduce the environment stochasticity caused by discretization error, future work will concentrate on deep reinforcement learning with function approximates. For providing stronger safety guarantees, we aim to implement a constrained reinforcement learning algorithm.

**Author Contributions:** Conceptualization, K.S., J.W., R.E. and M.d.W.; methodology, K.S.; software, K.S. and J.W.; writing—original draft preparation, K.S.; writing—review and editing, K.S., J.W. and M.d.W.; visualization, K.S.; supervision, R.E. and M.d.W.; funding acquisition, R.E. and M.d.W. All authors have read and agreed to the published version of the manuscript.

**Funding:** This work was executed with a Topsector Energy Grant from the Ministry of Economic affairs of The Netherlands.

**Institutional Review Board Statement:** Not applicable.

**Informed Consent Statement:** Not applicable.

**Data Availability Statement:** The code to reproduce the results presented in this study is released and can be accessed at [44].

**Conflicts of Interest:** The authors declare no conflict of interest.

## Nomenclature

| | |
|---|---|
| $\dot{m}^{max}$ | Maximum mass flow. |
| $\dot{m}_{c,t}$ | Mass flow at the time-step $t$ characterizing partial state space of a $Q$-learning algorithm $Q^{par}$— equal to the mass flow of the simulation environment $\widetilde{\dot{m}}_{s,t}$. |
| $\dot{m}_{s,t}$ | Mass flow in the supply network at the time-step $t$. |
| $\overline{F}_t$ | Upper bound on profit at the time-step $t$. |
| $\overline{R}^\tau$ | Upper bound on sub-reward functions concerning temperature (Hyperparameter of a $Q$-learning algorithm). |
| $\overline{R}^{h,under}$ | Upper bound on underdelivered heat sub-reward function (Hyperparameter of a $Q$-learning algorithm). |
| $\overline{R}^{profit}$ | Upper bound on profit sub-reward function (Hyperparameter of a $Q$-learning algorithm). |
| $\overline{R}^m$ | Upper bound on maximum mass flow sub-reward (Hyperparameter of a $Q$-learning algorithm). |
| $\underline{R}^\tau$ | Lower bound on sub-reward functions concerning temperature (Hyperparameter of a $Q$-learning algorithm). |
| $\underline{R}^{h,under}$ | Lower bound on underdelivered heat sub-reward function (Hyperparameter of a $Q$-learning algorithm). |
| $\underline{R}^{profit}$ | Lower bound on profit sub-reward function (Hyperparameter of a $Q$-learning algorithm). |
| $\underline{R}^m$ | Lower bound on maximum mass flow sub-reward (Hyperparameter of a $Q$-learning algorithm). |
| $\widetilde{\dot{m}}_{s,t}$ | Mass flow in the supply network at the time-step $t$ of the simulation environment. |
| $\widetilde{\dot{m}}_{s,t}^{sec}$ | Mass flow in the supply network of the secondary side grid at the time-step $t$ of the simulation environment. |
| $\widetilde{h}_t^c$ | Delivered heat to the consumer at the time-step $t$ of the simulation environment. |
| $A$ | Set of actions of $Q$-learning algorithm. |
| $a_0$ | Cost coefficient of the heat production. |
| $a_1$ | Cost coefficient of the electricity production. |
| $A_{sur}$ | Surface area of the pipe walls. |
| $A_p$ | Surface area of the pipe. |
| $a_t$ | Action of a $Q$-learning agent at the time-step $t$ (belongs to the set of actions $A$). |
| $C$ | Heat capacity of the water. |

| $c^{max}$ | Maximum electricity price. |
|---|---|
| $c^{min}$ | Minimum electricity price. |
| $c_t$ | Price of the electricity at the time-step $t$. |
| $F$ | Profit function over the optimization horizon $T$. |
| $F_t$ | Profit function at the time-step $t$. |
| $G_t$ | Cumulative reward at the time-step $t$. |
| $h$ | Thermal resistance. |
| $h_t^c$ | Delivered heat to the consumer at the time-step $t$. |
| $h^{c,max}$ | Maximal delivered heat to the consumer. |
| $h^{c,min}$ | Minimal delivered heat to the consumer. |
| $H_i$ | Heat corresponding to the $i$th characteristic point of the CHP operating region. |
| $h_t$ | Produced heat at the time-step $t$. |
| $L$ | Length of the pipe. |
| $m_{i,t}$ | Mass of the $i$th water chunk at the time-step $t$. |
| $N_\tau$ | Number of temperature points in a $Q$-learning state space. |
| $N_A$ | Number of action points of $Q$-learning algorithm. |
| $N_C$ | Number of water chunks. |
| $n_c$ | Number of water chunks of the same temperature. |
| $N_P$ | Set of characteristic points of the CHP unit. |
| $O$ | Set of observations of a $Q$-learning algorithm. |
| $o_t$ | Observation provided by a simulation environment at the time-step $t$ (belongs to the set of observations $O$). |
| $P$ | The probability of the transition (at the time-step $t$) to the state $s_{t+1}$ from the state $s_t$ under action $a_t$. |
| $P_i$ | Electricity corresponding to the $i$th characteristic point of the CHP operating region. |
| $p_t$ | Produced electricity at the time-step $t$. |
| $pr^{min}$ | Minimal heat load pressure at the heat exchange station. |
| $Q(s_t, a_t)$ | $Q$-value of executing action $a_t$ at the state $s_t$. |
| $Q^{full}$ | A full state space of a $Q$-learning algorithm. |
| $Q^{max}$ | Maximum heat demand. |
| $Q^{min}$ | Minimum heat demand. |
| $Q^{par}$ | A partial state space of a $Q$-learning algorithm. |
| $Q_t$ | Consumer's heat demand at the time-step $t$. |
| $R$ | Set of reward functions of $Q$-learning algorithm. |
| $R^{profit,max}$ | Maximum value of profit sub-reward function. |
| $R^{profit,min}$ | Minimum value of profit sub-reward function. |
| $R_t$ | Reward provided by a simulation environment at the time-step $t$ (belongs to the set of rewards $R$). |
| $R_t^{profit}$ | Profit sub-reward function of a $Q$-learning algorithm at the time-step $t$. |
| $R_t^{h,under}$ | Underdelivered sub-reward function of a $Q$-learning algorithm at the time-step $t$. |
| $R_t^{max,m}$ | Maximum mass flow sub-reward function of a $Q$-learning algorithm at the time-step $t$. |
| $R_t^{max,s,in}$ | Maximum supply network inlet temperature sub-reward function of a $Q$-learning algorithm at the time-step $t$. |
| $R_t^{min,r,in}$ | Minimum return network inlet temperature sub-reward function of a $Q$-learning algorithm at the time-step $t$. |
| $R_t^{min,s,in}$ | Minimum supply network inlet temperature sub-reward function of a $Q$-learning algorithm at the time-step $t$. |
| $S$ | Set of states of $Q$-learning algorithm. |
| $s_t$ | State of an environment of a $Q$-learning algorithm at the time-step $t$ (belongs to the set of states $S$). |
| $s_t^{ext}$ | External state space of a $Q$-learning algorithm at the time-step $t$. |
| $s_t^{int}$ | Internal state space of a $Q$-learning algorithm at the time-step $t$. |
| $s_t^{int,f}$ | Internal part of a $Q$-learning full state space at the time-step $t$. |
| $s_t^{int,p}$ | Internal part of a $Q$-learning partial state space at the time-step $t$. |
| $ss_t$ | Season at the time-step $t$—part of the state $s_t$ of a $Q$-learning algorithm. |
| $td_t$ | Time of the day at the time-step $t$—part of the state $s_t$ of a $Q$-learning algorithm. |
| $T$ | Optimization horizon. |

**Greek symbols**

| | |
|---|---|
| $\alpha$ | Learning rate of a $Q$-learning algorithm. |
| $\alpha_{t,i}$ | Variable representing the $i$th characteristic point of the CHP operating region at the time-step $t$. |
| $\Delta \dot{m}^{max}$ | Maximum mass flow violation. |
| $\Delta \dot{m}^{min}$ | Minimum mass flow violation. |
| $\Delta \dot{m}_c$ | Mass flow discretization step. |
| $\Delta \tau$ | Temperature discretization step. |
| $\Delta \tau^{max}$ | Maximum temperature violation. |
| $\Delta \tau^{min}$ | Minimum temperature violation. |
| $\Delta m_c$ | Mass discretization step. |
| $\Delta t$ | Optimization interval |
| $\epsilon$ | Exploration–exploitation parameter of a $Q$-learning algorithm. |
| $\gamma$ | Future rewards discount of a $Q$-learning algorithm. |
| $\rho$ | Density of the water. |
| $\tau_{env}$ | Temperature of the environment. |
| $\tau_{c_i,t}$ | Temperature of the $i$th water chunk at the time-step $t$. |
| $\tau_{r,t}^{in}$ | Temperature at the inlet of the return network at the time-step $t$. |
| $\tau_{r,t}^{out}$ | Temperature at the outlet of the return network at the time-step $t$. |
| $\tau_r^{hes,min}$ | Minimum temperature at the inlet of the return network. |
| $\tau_{s,t}^{out}$ | Temperature at the outlet of the supply network at the time-step $t$. |
| $\tau_{s,t}^{in}$ | Temperature at the inlet of the supply network at the time-step $t$. |
| $\tau_s^{max}$ | Maximum temperature at the inlet of the supply network. |
| $\tau_s^{min}$ | Minimum temperature at the inlet of the supply network. |
| $\tilde{\tau}_{r,t}^{out,sec}$ | Temperature at the outlet of the return network of the secondary side grid at the time-step $t$ of the simulation environment. |
| $\tilde{\tau}_{s,t}^{in,sec}$ | Temperature at the inlet of the supply network of the secondary side grid at the time-step $t$ of the simulation environment. |
| $\tilde{\tau}_{s,t}^{in}$ | Temperature at the inlet of the supply network at the time-step $t$ of the simulation environment. |
| $\tilde{\tau}_{s,t}^{out,sec}$ | Temperature at the outlet of the supply network of the secondary side grid at the time-step $t$ of the simulation environment. |
| $\tilde{\tau}_{s,t}^{out}$ | Temperature at the outlet of the supply network at the time-step $t$ of the simulation environment. |
| $\xi_\tau$ | Gradient of a temperature sub-reward function (Hyperparameter of a $Q$-learning algorithm). |
| $\xi_h$ | Gradient of a heat sub-reward function (Hyperparameter of a $Q$-learning algorithm). |
| $\xi_m$ | Gradient of a mass flow sub-reward function (Hyperparameter of a $Q$-learning algorithm). |
| $\xi_p$ | Gradient of a profit sub-reward function (Hyperparameter of a $Q$-learning algorithm). |

**Abbreviations**

| | |
|---|---|
| CHP | Combined heat and power plant |
| DHN | District heating network |
| HES | Heat exchange station |
| HS | Heat station |
| IPOPT | Interior point optimizer |
| LP | Linear program |
| MDP | Markov decision process |
| MILP | Mixed-integer linear program |
| MINLP | Mixed-integer nonlinear program |
| POMDP | Partial observable Markov decision process |
| $Q^{full}$ | Full state space $Q$-learning |
| $Q^{par}$ | Partial state space $Q$-learning |
| RES | Renewable energy source |
| RL | Reinforcement learning |
| UB | Upper bound |
| BS | Basic strategy |

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
