# Peer review of "Unlocking the Flexibility of District Heating Pipeline Energy Storage with Reinforcement Learning"

_energies, doi:10.3390/en15093290_

Round 1

Reviewer 1 Report

The paper deals about the integration of pipeline energy storage in the control of a district heating system through the use a reinforcement learning (RL) algorithm to estimate the dynamics of the system.

The paper covers an interesting topic that fits very well the scope of the journal.

The structure of the paper is clear and the language is proper.

The state of the art has been properly investigated.

What is the computational power required by the proposed approach with the respect to the state of the art?

What is the scalability of the proposed approach? Is it possible to optimize district heating system of an entire city using the proposed approach? What is the training time in this case?

Author Response

Thank you for providing valuable feedback on our manuscript. Please see the attachment.

Kind regards,

Ksenija Stepanovic

Reviewer 2 Report

Accept

Author Response

Thank you for providing valuable feedback on my manuscript.

Kind regards,

Ksenija Stepanovic

Reviewer 3 Report

The manuscript presents a study regarding the utilization of the pipelines' thermal inertia, considered as an energy storage system, to increase the flexibility in district heating networks.

The authors developed a reinforcement learning algorithm, and explores its potential in efficient, stable, safe, and real-time control.

The manuscript is well structured and documented.

The authors clearly describe the mathematical model, the optimization algorithm, the reinforcement learning algorithm, and the development of its components

The grid setting and parameter values and the simulation environment used for the reinforcement learning algorithm training and algorithms’ evaluation are also presented.

The authors analyzed the differences between using a full and a partial state space model for Q-learning. and benchmarked the best performing one against the state-of-the-art nonlinear programming approach.

Conclusions are supported by results.

Author Response

Thank you for providing valuable feedback on our manuscript.

Kind regards,
Ksenija Stepanovic

Reviewer 4 Report

The authors presented the possibility to use a reinforcement learning algorithm to estimate the system’s dynamics. The RL was adapted for pipeline energy storage considering temperature flow dynamics. The paper is actual and shows the authors contribution to the analysed topic. In general, it leaves a good impression. Only minor comments are given:

Abstract should be max of 200 words. Revise this part of the manuscript.

Quantities used in equations should be included in Nomenclature.

Figure 3: axis should have units.

Figure 7: text in sub-figures should be larger (presentation quality should be increased).

Add some quantitative results into Conclusions. It will help better understand your findings.

Author Response

(The authors gave the same response as above.)

Reviewer 5 Report

The work is interesting and contributes to the main body of knowledge. It shows the feasibility behind utilizsing the DH newtork as a therman energy storage for decoupling the thermal side from the electrical one for the CHP operation.

I would advise the atuhors to further elaborate on Figure 5 as it shows error bars with a wide range. Besides, it would be interersting if the authors can comment on the required volume for incoprporating a therman energy storage unit(s) in the DH in order to provide this flexibility. Moreover, how much saving can be achieved from using this strategy. Yet, it is still open whether such a configuration is technically feasible.

Best Regards

Author Response

(The authors gave the same response as above.)
